# Content-Based Image Copy Detection Using Convolutional Neural Network

**Xiaolong Liu [1,2,\*]**, **Jinchao Liang [1]**, **Zi-Yi Wang [3]**, **Yi-Te Tsai [4]**, **Chia-Chen Lin [3,5,\*]** and **Chih-Cheng Chen [6,\*]**

1 College of Computer and Information Science, Fujian Agriculture and Forestry University, Fuzhou 350002, China; j.c.liang1114@gmail.com
2 Digital Fujian Institute of Big Data for Agriculture and Forestry, Fujian Agriculture and Forestry University, Fuzhou 350002, China
3 Department of Computer Science and Information Engineering, National Chin-Yi University of Technology, Taichung 41170, Taiwan; g106035@gm.pu.edu.tw
4 Department of Computer Science and Communication Engineering, Providence University, Taichung 43301, Taiwan; yttsai@pu.edu.tw
5 Department of Computer Science and Information Management, Providence University, Taichung 43301, Taiwan
6 Department of Aeronautical Engineering, Chaoyang University of Technology, Taichung 413310, Taiwan
* Correspondence: xlliu@fafu.edu.cn (X.L.); ally.cclin@gmail.com (C.-C.L.); ccc@gm.cyut.edu.tw (C.-C.C.)

**Abstract:** With the rapid development of network technology, concerns pertaining to the enhancement of security and protection against violations of digital images have become critical over the past decade. In this paper, an image copy detection scheme based on the Inception convolutional neural network (CNN) model in deep learning is proposed. The image dataset is transferred by a number of image processing manipulations and the feature values in images are automatically extracted for learning and detecting the suspected unauthorized digital images. The experimental results show that the proposed scheme takes on an extraordinary role in the process of detecting duplicated images with rotation, scaling, and other content manipulations. Moreover, the mechanism of detecting duplicate images via a convolutional neural network model with different combinations of original images and manipulated images can improve the accuracy and efficiency of image copy detection compared with existing schemes.

**Keywords:** image copy detection; Inception; convolutional neural network; deep learning

## 1. Introduction

Multimedia forensics is one of the key technologies for digital evidence authentication in cybersecurity. The rapid expansion in the amount of digital content in social networks has also brought about a significant increase in the number of copyright infringements [1]. Nowadays, new image processing manipulations are rapidly developing and are incorporated into image processing software such as Photo Impact and Adobe Photoshop. Digital images are more likely to be copied and tampered while transmitting over the Internet. Therefore, concerns pertaining to the enhancement of security and protection against violations of digital images have become critical over the past decade [2]. Researchers are devoted to designing associated forensics algorithms, detecting the unauthorized manipulation, and then protecting the copyrights of original images.

In general, the current techniques for image copyright protection can be divided into digital watermarking [3–5] and content-based copy detection [6,7]. Digital watermarking is the mechanism that embeds digital watermarks as the copyright information into digital images, and the embedded

digital watermark is extracted as a basis for verification during copyright detection procedures. However, if there is a large number of images that need to be detected, the process of watermarking extraction verification would be time consuming and labor-intensive. On the other hand, content-based copy detection is the mechanism that captures unique feature values from original digital images, and could detect the suspected images from a large number of images based on the feature values. Digital watermarking and content-based copy detection techniques are considered to complement each other and could effectively protect the copyrights of digital images. It can be seen that content-based copy detection can first find a list of suspected images from a large number of images, and then the embedded watermark in suspected images can be extracted for further verification—in other words, when the image owner worries his/her images have been illegally manipulated and circulated over the Internet. S/he could first generate various manipulation images and feed them into the copy detection scheme for extracting image features. Next, s/he collects the similar images by using image search engine—i.e., Google Images. With the extracted image features, the image owner could filter out the suspicious images by using image copy detection. If the identified copy images are manipulated by the image owner herself/himself, s/he could exclude them based on the corresponding sources. As for the rest identified copy images, the image owner could claim her/his ownership and ask the unauthorized users to pay the penalty or take legal responsibility by extracting the hidden watermark.

The traditional image copy detection schemes [8] based on comparing the shape or texture in images can only detect the unauthorized copy of images. However, the infringement of digital images includes not only unauthorized copies, but also different image processing manipulations, such as rotation, compression, cropping and blurring, etc. Researchers observed that typical image manipulations may leave unique traces and designed forensic algorithms that extract features related to these traces and use them to detect targeted image manipulations [9,10]. However, most of the algorithms are designed to detect a single image processing manipulation. As a result, multiple tests must be run to authenticate an image, which would disturb the detection results and increase the overall false alarm rate among several detectors.

Lately, to effectively detect the infringed images with multiple image processing manipulations, content-based image copy detection schemes have been presented by scholars. Kim et al. [11] first applied the discrete cosine transform (DCT) to propose a novel content-based image copy detection mechanism, which can detect infringed digital images with scaling and other content modifications. However, it achieved a poor performance in the processing of rotated digital images, which can only detect the manipulation with a 180° rotation. To detect the digital images with different rotation angles, Wu et al. [12] proposed a scheme by extracting and comparing image feature values in the Y channel of the YUV (luminance, blue and red chrominance) color model, and could successfully detect the infringed images with rotation of 90° or 270°. However, it is still not able to effectively detect infringed images with others rotation angles. Lin et al. [13] proposed an image copy detection mechanism based on the feature extraction method of edge information, and successfully detected the infringed image with more different rotation angles. Lately, Zhou et al. [14] proposed a mechanism that can detect infringed images with any rotation angles by extracting and comparing two global feature values (gradient magnitude and direction) of the digital image. However, this mechanism requires experts to analyze the digital images in advance to extract effective feature values of the digital images.

Recently, a deep learning technique based on convolutional neural networks (CNNs) has taken on an extraordinary role in computer vision research fields, such as image recognition, object detection, and semantic segmentation [15]. It has been demonstrated that CNNs have an ability to automatically learn and classify the feature values of digital images, which is helpful to design associated forensics algorithms and detect the unauthorized manipulation for image copyright protection. Nowadays, several CNN models, such as Google Inception Net, ResNet, MobileNet, NASNet, and their different versions, have been developed by researchers to automatically learn classification features directly from data [16]. Inception V3 is now consider to be a representative version of the CNN family. Donahue et al.'s [17] research shows that the image feature extraction model in Inception V3 can retain

the parameters of all convolutional layers in the trained model and replace the last fully connected layer of the next training task. The target image may use the trained neural network to extract the features of the image. The extracted features will be used as input to solve the classification problem caused by the insufficient data of the neural network, which would also shorten the time required for training and improve the accuracy of classification.

In this paper, to study a high-performance convolutional neural network model for protection against violations of digital images by adaptively learning manipulation features, a content-based image copy detection scheme based on Inception V3 is proposed. The image dataset was transferred by a number of image processing manipulations, and the feature values were automatically extracted for learning and detecting the suspected unauthorized digital image. A Google Inception Net training model was used to automatically establish a convolutional layer to train the dataset. The performance under different training parameters is studied to design the optimum training model. The experimental results show that the proposed scheme takes an extraordinary role in detecting duplicate digital images under different manipulations, such as rotation, scaling, and other content modifications. The detection accuracy and efficiency are rapidly improved compared with other content-based image copy detection schemes.

The remainder of the paper is organized as follows: The related work of content-based image copy detection and the overview of CNNs are presented in Section 2. Section 3 provides a detailed description of the proposed scheme. The experimental results and comparison of related literature are presented in Section 4. Lastly, Section 5 concludes our work.

## 2. Related Works

### 2.1. Content-Based Image Copy Detection

Content-based copy detection applies the architecture of content-based image retrieval, which has higher requirements in recognition [18]. It is the technology that focuses on figuring out unauthorized images from the dataset that consists of suspected unauthorized digital images with rotation, scaling, and other content modifications. Researchers have focused on developing general-purpose image copy detection techniques to determine if an image has undergone malicious image processing manipulation [19]. Kim et al. [11] proposed the first representative content-based image copy detection mechanism based on DCT. The feature values of each digital image in the digital image database are generated based on an AC coefficient that is regarded as a sequential measure. To carry out this mechanism, the image is divided into 8*8 non-overlapping sub-image blocks, and an average intensity value is calculated for each block. The AC coefficients are determined from the sub-images, and sorted based on sequential measurement to represent the feature values. The experimental results shown that this mechanism can successfully detect images manipulated by equalization and contrast enhancement, as well as image processing such as watercolor, dynamic fuzzy mosaic, and Gaussian blur techniques. However, the performance is not satisfactory while dealing with rotation manipulated images.

To detect the images that are manipulated with different rotated angles, an improved content-based image copy detection scheme is proposed by Wu et al. [12]. They found that after the image was rotated, some of the edge features would move to other locations. In their proposed scheme, the picture is first converted into the YUV color model, and then only Y is taken as a plane to generate the feature value of the image. After dividing the digital image into an $8 \times 8$ size sub-image block, the average value of all the block is calculated. The correlation coefficient between each block and the surrounding eight blocks is extracted as the feature value. During the detection procedure, the image feature value in the Y channel of the YUV color model is extracted to detect the unauthorized images. The experimental results show that more image processing manipulation could be detected compared with Kim et al.'s scheme, even if the picture was rotated by 90° and 270°. However, the digital images that with more complicated rotation angles, such as 22.5° or 45°, are undetectable in this scheme.



Lin et al. [13] then proposed a mechanism based on feature values of the digital image edge information under different color models. In order to find each edge point, Lin et al.'s scheme scans each image with a $3 \times 3$ mask in zig-zag order. Their scheme can completely capture the edge features of the entire image, and can detect various rotation, shifting and cropping attacks compared with the image copy detection mechanisms proposed by Kim et al. and Wu et al. In order to detect the arbitrary rotation of the image, and at the same time achieve the desired performance in terms of accuracy and performance, Zhou et al. [14] proposed an efficient image copy detection scheme based on gradient magnitude and direction. In their method, the image is preprocessed to suppress the noise in order to improve the detection efficiency of the method. Secondly, an image segmentation strategy based on a pixel intensity sequence is used to divide the image into several rotation-invariant partitions. Finally, the features of the gradient amplitude and direction of the image are extracted, respectively, as the feature values of image. This scheme can detect the arbitrary rotation of the image. However, it requires experts to analyze the digital images in advance to find effective feature values of the digital images.

Although the schemes mentioned above have resulted in significant gains in accuracy of detecting unauthorized images that have undergone image processing manipulations, the specific image feature values should be manually extracted for detection, which is time consuming. Deep learning approaches, particularly CNNs that can provide their ability to automatically learn classification model directly from data, exploit a new area of study to improve detection accuracy and efficiency by automatically detecting whether the query image is an unauthorized duplicated image once it is input into the detection model.

### 2.2. Convolutional Neural Network

A CNN [20] is a deep learning model constructed by referring to the visual nerve of neural networks. Its main function is to let the computer find out the correlation between data and information on its own, and extract useful features to replace manually defined features. It is a kind of feedforward neural network, whose artificial neurons can respond to a part of the surrounding units in the coverage area and meanwhile preserve the features in the scope, which exhibits an excellent performance for a large number of image processing methods. Figure 1 shows a representative structure of CNN [21], which usually contains a convolutional layer, pooling layer, and fully connected layer.

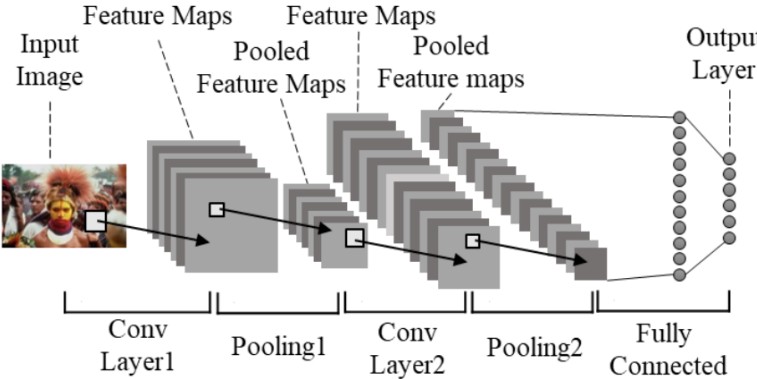

**Figure 1.** The representative convolutional neural network (CNN) structure [21].

The convolution layer extracts the feature values of inputs and performs a convolution operation between the preprocessed image and the specific filter. The filter coefficients in each layer are initially seeded with random values, then learned using back-propagation algorithms [22,23]. The set of convolutional layers yields a large volume of feature maps. The pooling layer takes an extraordinary role in reducing the dimensionality of these features, compressing data volume and decaying overfitting. Max pooling is one of the common pooling operations that divides the input digital image into several rectangular areas and figures out the maximum value of each rectangular area. The fully connected

layer automatically identifies the target based on the feature maps trained by the processing of multiple convolutional and pooling layers with the weight and connection strength.

Recently, CNNs have been successfully used on computer vision fields, such as image recognition, object detection, and semantic segmentation. Bayar et al. [24] developed an innovative convolutional layer designed for suppressing image content that is adapted to learn for manipulating detect feature values. They selected convolution filtering to establish the first layer of a convolutional since the higher layers of the CNN take an extraordinary role in learning appropriate methods for extracting low-dimensional detection features from high-dimensional prediction errors. The images in the dataset were divided into 256 × 256 blocks, and a total of 87,000 blocks were applied for model training, shortening the model training time within one hour. The results demonstrated that CNNs take on an extraordinary role in terms of detection efficiency compared to traditional schemes. Yan et al. [25] rapidly improved the accuracy of digital image detection based on changing the feature values. It put the image dataset into a CNN model to extract the feature values, and optimized the feature values based on a Bag-of-Words model. Their scheme improved the accuracy of digital image detection, and the accuracy rate reached up to 94.59%. Cozzolino et al. [26] transferred training images out of multiple processing images of the same type (for example, gaussian noise) for repeated training to improve the stability of the deep neural network, and the accuracy rate was up to 93.9% based on the original Inception training model. Zheng et al. [27] converted the digital image into two color modes (grayscale and color), stitching the color images from top to bottom, and establishing a convolutional neural network model to, respectively, learn the feature values of grayscale images and color images. Their scheme combined two outputs via fully connected layers, and extracted feature values from a pseudo-Siamese model to keep the stitched image to maintain the feature values of the two original images. It could not only prevent the brightness features from affecting the identification, but also find the most recognizable features from the color images.

Nowadays, several CNN models, such as Google Inception Net, ResNet, MobileNet, NASNet, and their different versions, have been developed in various field of engineering for pattern recognition, image classification, and text categorization [28–30] as well as in many traffic safety-related research [31,32]. Researchers no longer need to spend a lot of time to design effective feature value alignment between the original image and the query image, which can accelerate the study of image copy detection. To gain ideal training results, a training model of CNNs performs an extraordinary role in conducting machine learning for the relationship between data information and specimens in the dataset. Therefore, in this work we will use a CNN as the basic model to effectively find out the list of suspected unauthorized images and improve the performance of intellectual property protection. To see whether CNNs can demonstrate outstanding detection performance on coped images, we collect 43 image processing manipulations which are commonly seen in the image processing applications to generate copied images. Moreover, to present more insight of CNNs on image copy detection, in this work, not only the detection accuracy under different training models such as Inception_v3, ResNet_v2, MobileNet_v2, NASNet_large will be presented, but different datasets used for training will also be explored.

## 3. Methodology

In this study, the conventional neural network (CNN) method was adopted to develop image copy detection based on the training of the detection CNN model to determine whether the query digital image is a suspected unauthorized image. The image copy detection model of this scheme consists of image preprocessing and detection model training procedures. To gain an expanded digital image dataset, images were first converted into 44 different forms including the original image during image preprocessing. After that, the digital images were divided into two sorts of datasets by selecting 70% of the digital images as the training dataset, and the remaining 30% of the digital images are regarded as the test dataset. The image copy detection model was then trained by automatically extracting feature values of each image based on Inception V3. Finally, the detection model is used to detect whether the

query image is a suspected unauthorized image or not. The flowchart of the proposed image copy detection scheme is shown in Figure 2.

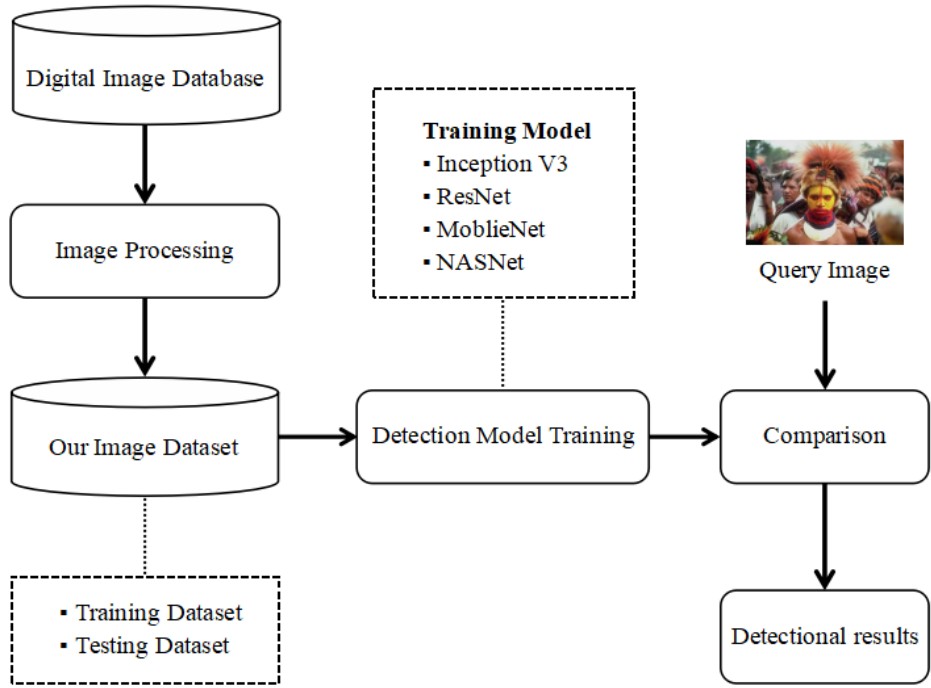

**Figure 2.** Flowchart of the proposed image copy detection scheme.

*3.1. Image Preprocessing*

At the image preprocessing stage, various common image processing manipulations, such as translation, rotation, and other content modifications, were firstly performed on each image via image processing software to generate 44 different forms of the image that are regarded as manipulated digital images. It was noted that the original image and its corresponding manipulation images will form a group—in other words, they will have the same label. This is because our proposed image copy detection scheme aimed to shrink the training time and scale of the training dataset. When the image owner wants to verify whether the similar images collected from the Internet contains her/his image, they only needs to generate various manipulation images in advance and feed them into our proposed image copy detection scheme for training the features of her/his image. Then, the image copy detection scheme would identify the most suspicious image from the similar images. To prevent overfitting, 70% of the images which belong to the same group were selected and finally formed a training dataset and 10% of the training dataset were randomly extracted for verification during model training. The remaining 30% of the images were regarded as the test dataset.

*3.2. Detection Model Training*

Tensorflow was selected as the development environment to train the dataset, and the Google Net Inception V3 [33] convolutional neural network architecture was utilized to learn the training model, which contains four significant parts—i.e., $1 \times 1$ convolution, $3 \times 3$ convolution, $5 \times 5$ convolution, and $3 \times 3$ maximum pooling, as shown in Figure 3.

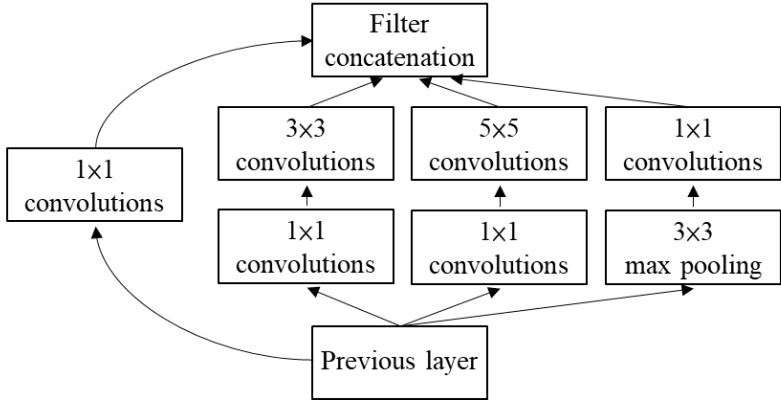

**Figure 3.** Google Net Inception Module.

Christian Szegedy et al. [33] further explored different Inception models—Figures 4b and 5a,b based on Figure 4a.

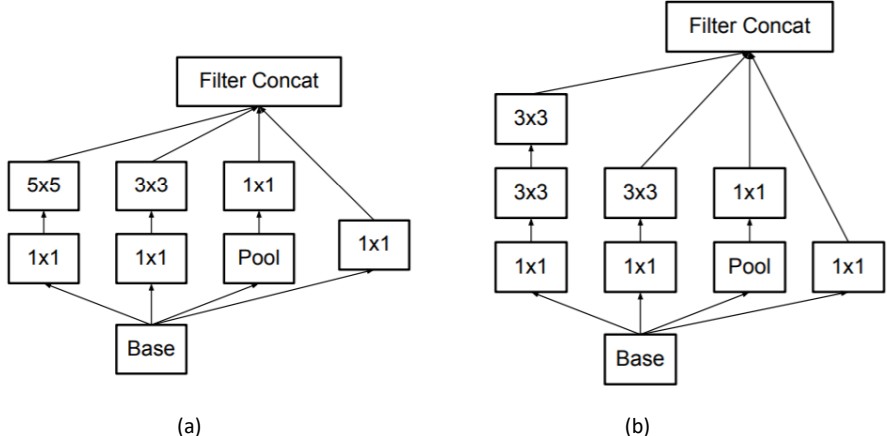

(a)                                                                 (b)

**Figure 4.** (**a**) Original Inception module as described in [33]; (**b**) Inception modules where each $5 \times 5$ convolution is replaced by two $3 \times 3$ convolutions.

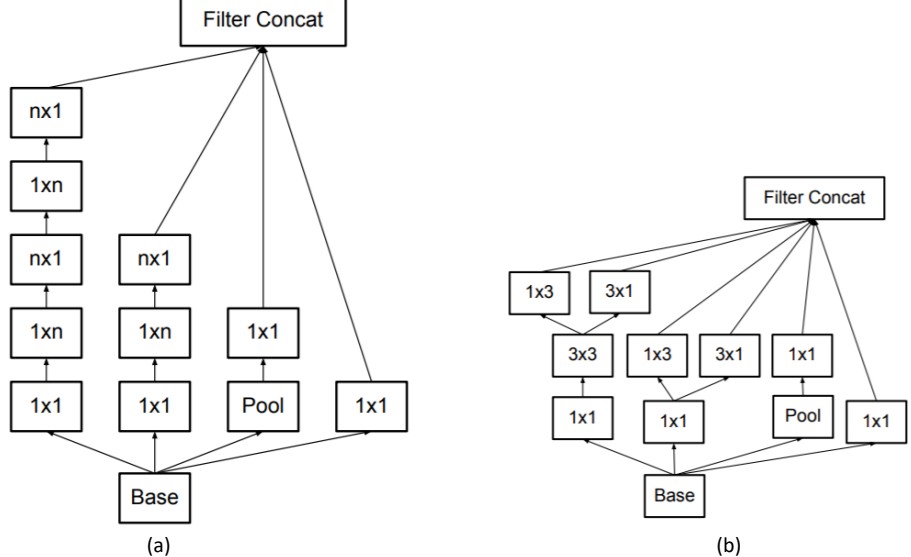

(a)                                                                 (b)

**Figure 5.** (**a**) Inception modules after the factorization of the $n \times n$. convolutions. Here, $n = 7$; (**b**) Inception modules with expanded the filter bank outputs.

To gain optimized feature values, different information of digital images was extracted by multiple convolutional layers and combining the outputs of these convolutional layers as the input of the next layer [34]. Inception V3 improves the idea of factorization that decomposes a relatively large two-dimensional convolutional layer into petty small one-dimensional convolutional layers to accelerate the computing speed and deepness of the network. The convolution structure performs a significant role in handling spatial feature values and increases the diversity of feature values. The layout of our network is shown in Table 1. The output size of each module is the input size of the next one. The variations of the reduction technique are depicted in [33] and are also used in our proposed network architecture to increase the accuracy while maintaining a constant amount of computation complexity. The classifier in our network is set as softmax. It is noted that both sigmoid and softmax can be used for multiclass classification. However, the sigmoid looks at each raw output value separately. In contrast, the outputs of softmax are all interrelated. Based on the scenario of image copy detection, softmax is selected as classifier in our proposed scheme.

**Table 1.** The outline of our proposed network architecture.

| Types | Patch Sizes/Strides | Input Sizes |
|---|---|---|
| conv | $3 \times 3/2$ | $299 \times 299 \times 3$ |
| conv | $3 \times 3/1$ | $149 \times 149 \times 32$ |
| conv padded | $3 \times 3/1$ | $147 \times 147 \times 32$ |
| pool | $3 \times 3/2$ | $147 \times 147 \times 64$ |
| conv | $3 \times 3/1$ | $73 \times 73 \times 64$ |
| conv | $3 \times 3/1$ | $73 \times 73 \times 64$ |
| pool | $3 \times 3/2$ | $71 \times 71 \times 192$ |
| conv | $3 \times 3/1$ | $35 \times 35 \times 256$ |
| 2× Inception | Figure 5b | $35 \times 35 \times 288$ |
| 5× Inception | Figure 5a | $17 \times 17 \times 768$ |
| 3× Inception | Figure 4b | $8 \times 8 \times 1280$ |
| pool | $8 \times 8$ | $8 \times 8 \times 2048$ |
| softmax | classifier | $1 \times 1 \times 2048$ |

To determine the relationship between images and labels, the preprocessed digital image dataset with corresponding labels is put into the model for training. The detection model training consists of two stages: (a) training parameters setting and (b) model accuracy detection. During the setting training parameters stage, the training amplitude, time, and size of the input digital image dataset are adjusted, and the size of the feature map and the number of extracted feature values are also determined.

In the Inception training model, the input size of the digital image was first set and divided into three depths based on the three channels of RGB as the input layer. The convolution layer depth was set to 32, which will gain 32 filters with a depth, height, and width of 3. The convolution results are 32 feature values with heights and widths of 149 based on the following equations:

$$W_2 = (W_1 - F + 2P)/S + 1 \qquad (1)$$

$$H_2 = (H_1 - F + 2P)/S + 1 \qquad (2)$$

where $W_1$ is the width of the unconvoluted digital image; $W_2$ is the width of the convoluted feature map; $F$ is the width of the filter; $P$ is the number of zeros padded around the original digital image; $S$ is the stride; $H_2$ is the height of the convoluted feature map; $H_1$ is the height of the unconvoluted digital image.

Besides image size, training steps and learning rate can also be set as the parameters to justify the training results. The default training steps were set as 4000; this is because more training steps may improve the training accuracy but lower validation accuracy. The learning rate controls the training level of the final layer, and the default value was set as 0.01. After each training stage, the accuracy of

the model was checked by the verification dataset. The training stage was repeated until the model accuracy reached the setting value (90%), as shown in Figure 6.

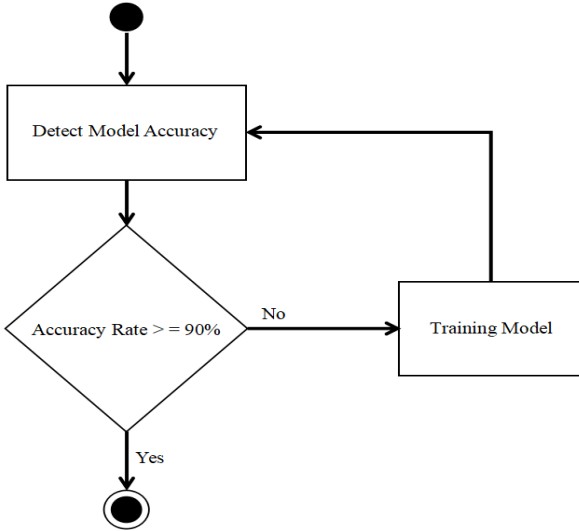

**Figure 6.** Model accuracy detection.

During the image copy detection procedure, the system will automatically find five images in the dataset that are most similar to the query image as the suspected image list. The suspected image list is then used to compare and verify whether the query image is a suspected unauthorized image or not.

## 4. Results

To train the effectiveness and accuracy of the proposed method, the WBIIS database [35,36], which contains 10,000 open-source and copyright-free digital images in jpeg format with sizes of $384 \times 256$ and $256 \times 384$, was chosen as the test database in the experiment; the example of the test images is shown in Figure 7. We selected 70% of the digital images as the training dataset and the remaining 30% of the digital images as the test dataset. To train whether the proposed scheme can detect unauthorized duplicated images under various attacks, each image was transferred into different forms based on 43 image processing manipulations, as shown in Figure 8. It is noted that the watermarked image was not included in manipulations images as shown in Figure 8. This is because we assumed that once the original image carries the hidden watermark, its corresponding manipulations will also carry the hidden watermark. Therefore, we only need to focus on discussing whether our proposed copy detection with CNNs could identify the manipulations or not.

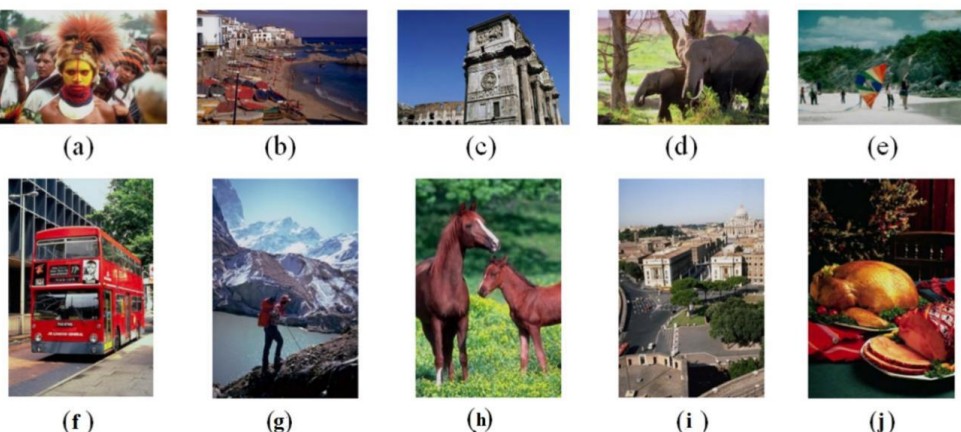

**Figure 7.** Example of the test images.

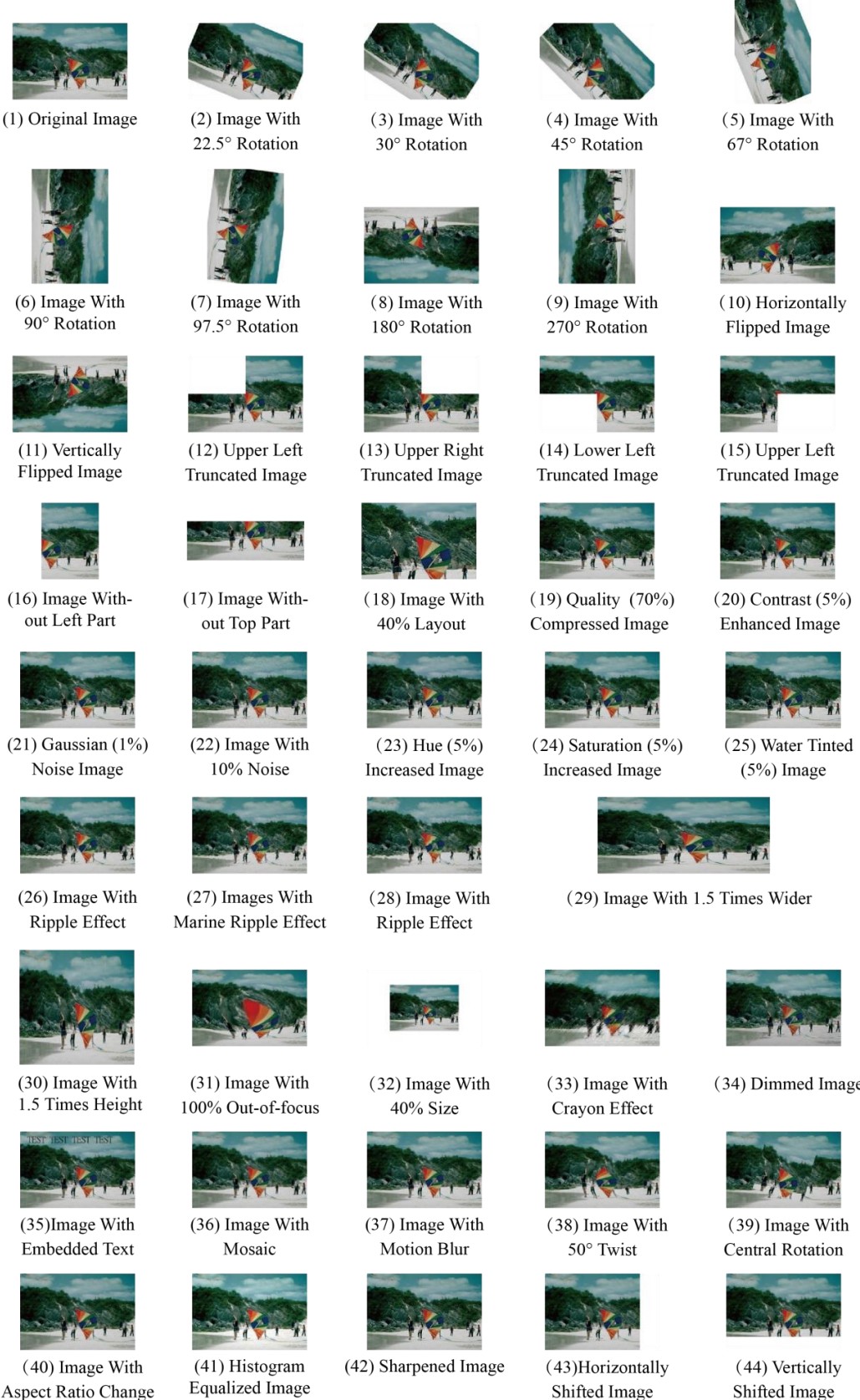

**Figure 8.** Digital images with different image processing manipulations.

*4.1. Results in Different Learning Rates*

To study the relationship between the learning rate and the training results, this scheme set the initial value of the learning rate parameters of the trained Inception V3 image feature vector model to 0.005, 0.01, 0.5, and 1.0 to compare the training results, as shown in Table 2. Table 2 indicates that the result of training time is inversely proportional to the learning rate. It also shows that the training time and accuracy of the default learning rate (i.e., 0.01) are satisfactory—431 and 96.23%, respectively. The training time will increase and the accuracy will decrease if the learning rate is decreased to 0.005. Therefore, we speculate that while the initial value of the learning rate parameter is less than 0.01, the longer training time will lead to overfitting and decreases the accuracy. To obtain a higher accuracy, the initial value of the learning rate should be adjusted upward, and the accuracy will reach up to 99.55% while the learning rate is 1.0.

**Table 2.** The training results (time and accuracy) under different learning rates.

| Learning Rate | Training Time | Accuracy |
|---|---|---|
| 0.005 | 436 s | 95.01% |
| 0.01 | 431 s | 96.23% |
| 0.1 | 320 s | 98.29% |
| 0.5 | 319 s | 97.73% |
| 1.0 | 315 s | 99.55% |

*4.2. Comparison Results in Different Training Models*

To compare the results of Inception_v3 training model in Tensorflow with other training models, we future trained the detection model in ResNet_v2, MobileNet_v2, and NASNet large; the comparison results are shown in Table 3. The results show that all of the training models can achieve very high accuracy results, whereas the training time varies widely. The smallest neural network MobileNet_v2 has the shortest training time, and the accuracy rate can reach up to 98.77%. Although the accuracy rate of NASNet_large is 99.34%, the training time is also the longest. This is because NASNet_large not only trains for the most effective feature values, but also trains to gain the best neural network architecture, which would result in more training time. With respect to Inception_v3, the accuracy is 99.55% which is the best in the four models when the learning rate is set as 1, and the training time (315 s) is also very significant. Therefore, in general, the comparison results demonstrate that the Inception_v3 model used in the proposed scheme can achieve the best performance compared with other models.

**Table 3.** Comparison of the results in different training models.

| Training Model | Training Time | Accuracy |
|---|---|---|
| Inception_v3 | 315 s | 99.55% |
| ResNet_v2 | 422 s | 98.62% |
| MobileNet_v2 | 292 s | 98.77% |
| NASNet large | 1269 s | 99.34% |

*4.3. Comparison Results in Different Training Datasets*

Certainly, as the results in Table 3 demonstrate, with all manipulated images including the original image serve as the training dataset, our proposed method works with Inception_v3 and can offer a detection accuracy of up to 99.55%. However, it is impractical because such accuracy is dependent on all manipulated images being included in the training dataset. Therefore, the second experiment was conducted which only uses original images in the training dataset except for 43 manipulated images. Unfortunately, the detection accuracy rate is then decreased to 90.06%. Under such circumstances, we can see that 7 of the 10 test images with the Crayon effect cannot be detected, as is the case for six rotation manipulations, such as 22.5°, 45°, 67°, 90°, 180°, and 270° rotations. The number of

undetected manipulated images is significantly decreased, and the worst case is that 4 of 6 rotation manipulated images are undetectable. For 5 of the 10 test images, the amount of undetectable rotation manipulated images ranges from 3 to 4. To increase the detection accuracy rate, the third experiment was conducted, in which the test dataset including original images and manipulated images with 45° rotation was used. With the assistance of the manipulated images with a 45° rotation, the CNN obtains extra features regarding the rotation manipulation. It is noted that the detection accuracy rate changed to 96.47%, which is 6% more than that trained with the dataset excluding manipulated images.

Based on the above experimental data demonstrated in Table 4, we can see that the supplementary features offered by manipulated images with a 45° rotation significantly increase the detection accuracy rate of our proposed scheme. There are 8 of 10 test images which have their manipulated images with various rotations successfully detected. As for manipulated images with the "Crayon effect", there are 4 of 10 test images that cannot be correctly detected. However, the corresponding similarities among the manipulated images with the Crayon effect and their query images are significantly increased. After carefully observing the corresponding test images which cannot be identified from the manipulated images with the "Crayon effect", it is found that this only occurs when test images are complex and contain vivid colors.

**Table 4.** Comparison of the results in different training datasets.

| Training Datasets | Accuracy |
| --- | --- |
| Dataset including all manipulated images | 99.55% |
| Dataset only including original images | 90.06% |
| Dataset including original images and images with 45° rotation | 96.47% |

### 4.4. Comparison with Content-Based Image Copy Detection Schemes

To detect the performance and accuracy of the proposed scheme, the image copy detection results are compared with some representative content-based image copy detection schemes—i.e., Lin et al.'s scheme [13], Wu et al.'s scheme [12], and Kim et al.'s scheme [11]. We randomly selected 10 images from the image database, and each image was processed based on the 44 image processing manipulations to generate the query digital images. We calculated the number of images that can be detected in 10 images of each image processing manipulation. Only if all of the 10 images in each image processing manipulation are detected will the detection result be marked as "Yes", otherwise the result will be marked as "No". Table 5 shows the detection results of the proposed scheme and other content-based image copy detection schemes in different image processing manipulations.

In the experiment, the proposed image copy detection scheme with the training dataset including all manipulated images could successfully detect all query digital images with an accuracy of 100%, whereas the accuracy of Lin et al.'s, Wu et al.'s, and Kim et al.'s schemes can only achieve 88.6, 70.5 and 63.6%, respectively. In terms of the robustness of detection schemes, the proposed scheme can resist all of 44 image processing manipulations which is the best in all of the four schemes. Lin et al.'s scheme can resist most of the image processing manipulations, expect for the "45° Rotation", "Crayon Effect", "Mosaic", "Central Rotation", and "Sharpened". The performance of Wu et al.'s and Kim et al.'s schemes is not satisfactory while dealing with such image processing manipulations. With respect to detection time, Lin et al.'s, Wu et al.'s, and Kim et al.'s schemes should manually extract specific image feature values for detection, which is time consuming. On the contrary, the proposed scheme would automatically detect whether the query image is an unauthorized duplicated image once it is input into the detection model. Therefore, the comparison results demonstrate that the proposed scheme outperforms the compared image copy detection schemes in terms of detection time and accuracy. Once the proposed image copy detection scheme with the training dataset included original images and manipulated images with 45° rotation, the detection accuracy rate is the same as that of Lin et al.'s scheme. Although, both of the schemes cannot deal with manipulated images with the Crayon effect. Our proposed scheme still can identify 4 of 10 test images. As for manipulated images with 67°, 90°, 270°, and 180° rotations, the average detection accuracy rate still remains

about 80%. Moreover, our proposed scheme can resist "45° Rotation", "Mosaic" "Central Rotation", and "Sharpened", which cannot be handled by Lin et al.'s scheme.

**Table 5.** Comparison I of the detection results in different image processing manipulations.

| Image Processing Manipulation | Proposed Scheme with Dataset Including Original Images and Images with 45° Rotation | Proposed Scheme with Dataset Including All Manipulated Images | Lin et al.'s Scheme | Wang et al.'s Scheme | Kim et al.'s Scheme |
|---|---|---|---|---|---|
| 22.5° Rotation | Yes | Yes | Yes | No | No |
| 45° Rotation | Yes | Yes | No (10/10) | No | No |
| 67° Rotation | No (2/10) | Yes | Yes | No | No |
| 90° Rotation | No (3/10) | Yes | Yes | Yes | No |
| 270° Rotation | No (2/10) | Yes | Yes | Yes | No |
| 180° Rotation | No (1/10) | Yes | Yes | Yes | Yes |
| Upper Left Truncated | Yes | Yes | Yes | No | No |
| Upper Right Truncated | Yes | Yes | Yes | No | No |
| Lower Left Truncated | Yes | Yes | Yes | No | No |
| Lower Right Truncated | Yes | Yes | Yes | No | No |
| Crayon Effect | No (4/10) | Yes | No (10/10) | Yes | Yes |
| Mosaic | Yes | Yes | No (10/10) | Yes | Yes |
| Twisted | Yes | Yes | Yes | Yes | No |
| Central Rotation | Yes | Yes | No (10/10) | No | No |
| Aspect Ratio Change | Yes | Yes | Yes | No | No |
| Histogram Equalized | Yes | Yes | Yes | No | No |
| Sharpened | Yes | Yes | No (10/10) | No | No |
| Horizontally Shifted | Yes | Yes | Yes | No | No |
| Vertically Shifted | Yes | Yes | Yes | No | No |
| The rest 25 manipulations | Yes | Yes | Yes | Yes | Yes |
| Detection Number | 39 | 44 | 39 | 31 | 28 |
| Detection Rate | 88.6% | 100% | 88.6% | 70.5% | 63.6% |

Table 6 demonstrates Comparison II when images with 45° rotation/90° rotation are added to the training dataset. Here, we can see the detection performance of our proposed scheme on manipulation with different rotations angles increased, although the "Crayon Effect" manipulation is still the weakness of our proposed scheme. After carefully observing the "Crayon Effect" manipulation, we found it leads to the detection capability being decreased when the texture of the image is more blurred.

**Table 6.** Comparison II of the detection results in different image processing manipulations.

| Image Processing Manipulation | Proposed Scheme with Dataset Including Original Images and Images with 45° Rotation/90° Rotation | Proposed Scheme with Dataset Including All Manipulated Images | Lin et al.'s Scheme | Wang et al.'s Scheme | Kim et al.'s Scheme |
|---|---|---|---|---|---|
| 22.5° Rotation | Yes | Yes | Yes | No | No |
| 45° Rotation | Yes | Yes | No (10/10) | No | No |
| 67° Rotation | No (1/10) | Yes | Yes | No | No |
| 90° Rotation | Yes | Yes | Yes | Yes | No |
| 270° Rotation | Yes | Yes | Yes | Yes | No |
| 180° Rotation | Yes | Yes | Yes | Yes | Yes |
| Upper Left Truncated | Yes | Yes | Yes | No | No |
| Upper Right Truncated | Yes | Yes | Yes | No | No |
| Lower Left Truncated | Yes | Yes | Yes | No | No |
| Lower Right Truncated | Yes | Yes | Yes | No | No |
| Crayon Effect | No (4/10) | Yes | No (10/10) | Yes | Yes |
| Mosaic | Yes | Yes | No (10/10) | Yes | Yes |
| Twisted | Yes | Yes | Yes | Yes | No |
| Central Rotation | Yes | Yes | No (10/10) | No | No |
| Aspect Ratio Change | Yes | Yes | Yes | No | No |
| Histogram Equalized | Yes | Yes | Yes | No | No |
| Sharpened | Yes | Yes | No (10/10) | No | No |
| Horizontally Shifted | Yes | Yes | Yes | No | No |
| Vertically Shifted | Yes | Yes | Yes | No | No |
| The rest 25 manipulations | Yes | Yes | Yes | Yes | Yes |
| Detection Number | 42 | 44 | 39 | 31 | 28 |
| Detection Rate | 95.45% | 100% | 88.6% | 70.5% | 63.6% |

To further evaluate whether our proposed image copy detection with a CNN successfully learns the feature of training set images, and then could identify some manipulated images which are slightly different from those in the training set, five extra manipulated images were generated as 10° rotation, image with 15% noise, image twisted 25°, image 1.4 times wider, and horizontally shifted image, and added into query images.

From the accuracies listed in Table 7, we can see the accuracies of the training set are nearly the same as previous experimental results. As for the accuracies of the testing set, they are slightly lower than those of accuracies of the training set. We believe this is mainly caused by either the "Crayon Effect" or few angle rotations manipulations. Such results are consistent with previous experiments and prove that our proposed image copy detection with a CNN could take the advantage of CNNs to learn the features of images so that some manipulations can still be identified even if they are out of the scope of the training set.

**Table 7.** Comparison of the results in different training datasets.

| Training Datasets | Accuracy of Training | Accuracy of Testing |
|---|---|---|
| Dataset including all manipulated images | 99.54% | 98.35% |
| Dataset only including original images | 90.07% | 85.08% |
| Dataset including original images and images with 45° rotation | 96.38% | 88.2% |
| Dataset including original images and images with 45°/90° rotation | 98.12% | 94.6% |

## 5. Conclusions

An image copy detection scheme based on the Inception V3 convolutional neural network is proposed in this paper. The image dataset is firstly transferred by a number of image processing manipulations for training the detection model with feature values. During the image copy detection procedure, the system will automatically find out the suspected image list to compare and verify whether the query image is a suspected unauthorized image or not. The experimental results show that the training accuracy will reach up to 99.55% while the training dataset including all manipulated images and the learning rate is set as 1.0, which is superior to the model of ResNet_v2, MobileNet_v2, and NASNet large. Even though the training dataset only includes original images and manipulated images with 45° rotations, our proposed scheme still outperforms Lin et al.'s scheme on the amount of manipulation types which have been successfully identified. Certainly, 80% of the manipulated images with rotations can be detected, but 40% of the manipulated images with the Crayon effect are hard to identify when the training dataset only contains original images and manipulated images with 45° rotation with our proposed scheme. Such a result indicates the lowest requirement of the training set with our proposed scheme. However, the experimental results also pointed out that our proposed image copy detection scheme can identify some manipulations which are slightly different from those in the training set. We believe it is a benefit that deep learning can effectively extract the supplementary features of images in the training set. Certainly, with experiments conducted in this work, we also discovered that the additional information is still limited by feeding manipulated images with 45° rotations. In the future, we will try to explore how to improve the detection performance of CNNs on the "Crayon Effect" and find out the tradeoff between the amount of manipulation types which should be included in the training set and the accuracy of the testing set to increase the practicability of the image copy detection scheme.

**Author Contributions:** Conceptualization, X.L. and C.-C.L.; Data curation, Z.-Y.W.; Formal analysis, C.-C.L.; Funding acquisition, X.L. and C.-C.L.; Investigation, Z.-Y.W. and C.-C.C.; Methodology, X.L., Z.-Y.W., and C.-C.L.; Project administration, C.-C.L.; Resources, C.-C.L.; Software, Z.-Y.W. and J.L.; Supervision, X.L., Y.-T.T. and C.-C.L.; Validation, J.L.; Writing—original draft, X.L., Z.-Y.W., and J.L.; Writing—review and editing, and C.-C.L. All authors have read and agreed to the published version of the manuscript.

**Funding:** This work was supported by the fund of National Natural Science Foundation of China (Grants No. 61702102), Natural Science Foundation of Fujian Province, China (Grant No. 2018J05100), Foundation for Distinguished Young Scholars of Fujian Agriculture and Forestry University (Grant No. xjq201809).

**Conflicts of Interest:** The authors declare no conflict of interest. The funders had no role in the design of the study; in the collection, analyses, or interpretation of data; in the writing of the manuscript, or in the decision to publish the results.

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
