# Peer review of "Content-Based Image Copy Detection Using Convolutional Neural Network"

_electronics, doi:10.3390/electronics9122029_

Round 1

Reviewer 1 Report

The paper is relevant and the work described is very interesting. The authors present an image copy detection scheme based on Inception-V3 convolutional neural network. Experiments demonstrate good performance.  However the research design could be improved. The authors should provide more justification about the selection of the training models, detection model and learning rate. The language also needs improvement. There are grammar errors in places.

Author Response

Journal: Electronics (ISSN 2079-9292)

Manuscript ID: electronics-996564

Type: Article   Number of Pages: 19

Title: Content-based image copy detection using Inception convolutional neural network

Authors: Xiaolong Liu, Jinchao Liang, Zi-Yi Wang, Yi-Te Tsai , Chia-Chen Lin *, Chih-Cheng Chen *

Dear Editor,

Thank you very much for your letter and for the comments addressed by the reviewers. These comments are very valuable and helpful for our paper. We appreciate the careful, constructive, and generally favorable reviews given to our paper by the reviewers.

We believe we have adequately addressed all the excellent advices and questions raised by reviewers. Furthermore, we checked the manuscript and made sure the submitted manuscript is correct.

Please contact us if any further questions remain.

Sincerely yours,

Prof.  Chih-Cheng Chen 

Response to the comments of reviewers:

Reviewer 1:

Comments and Suggestions for Authors

Q1: The paper is relevant and the work described is very interesting. The authors present an image copy detection scheme based on Inception-V3 convolutional neural network. Experiments demonstrate good performance.  However, the research design could be improved. The authors should provide more justification about the selection of the training models, detection model and learning rate.

Response: Many thanks for your valuable comments. In general, both sigmoid and softmax can be used for multiclass classification. However, the sigmoid looks at each raw output value separately. In contrast, the outputs of a softmax are all interrelated. In the scenario of image copy detection scheme, when the image owner wants to verify whether the similar images collected from the Internet contain her/his image. Image owners need to generate various manipulation images in advance and feed them into our proposed image copy detection scheme for training the features of her/his image. Then, the image copy detection scheme would identify the most suspicious image from similar images. Based on the above scenario, softmax is selected as a classifier in our work. To comply with the referee’s comment, the related sentences have been added one line from 281 to 284 in this revision. Moreover, the descriptions regarding our layout of our network, training model, and learning rates have been added in this revision. Please refer to the lines from 277 to 285, lines from 300 to 307 in this revision.

Q2. The language also needs improvement. There are grammar errors in places.

Response: Many thanks for your valuable comments. We have carefully checked through our manuscript when we preparing our revision. We sincerely hope the improved revision meets your requirement.

Reviewer 2 Report

In this work, the authors investigate the development of a novel approach to detecting duplication of copyright images using a convolutional neural network architecture. While the problem is of interest, there are a few concerns regarding the soundness of the presented solution for solving the problem that must be addressed.

It seems to the reviewer that the developed approach is a manipulation detector as opposed to a unauthorized copyrighted image detector. How do the authors discriminate between an authorized/copyrighted image that has one of these effects applied to it vs. a manipulation on the original image. For example, if the original copyrighted image is vertically flipped and the manipulation was to put it right side up, would this approach detect this manipulation? The authors need to better explain their methodology between what is considered an authorized image vs. an unauthorized image.

Other comments:

  • It would be helpful to give a direct example of the need for copyright detection technology. What is a real world problem that this solution would help solve?
  • Watermarks are discussed as one primary way to copyright an image but wasn't considered in the permutation set by the authors.
  • How did the authors balance the training dataset between unauthorized image manipulations and the original images to avoid training bias?
  • Doesn't seem like the authors utilized the verification dataset to avoid overfitting (e.g. through the use of 'patience' or 'early stopping' mechanisms)
  • Very hard for the reader to see some of the manipulations shown in Figure 6.

Author Response

Journal: Electronics (ISSN 2079-9292)

Manuscript ID: electronics-996564

Type: Article   Number of Pages: 19

Title: Content-based image copy detection using Inception convolutional neural network

Authors: Xiaolong Liu, Jinchao Liang, Zi-Yi Wang , Yi-Te Tsai , Chia-Chen Lin * , Chih-Cheng Chen *

Dear Editor,

Thank you very much for your letter and for the comments addressed by the reviewers. These comments are very valuable and helpful for our paper. We appreciate the careful, constructive, and generally favorable reviews given to our paper by the reviewers.

We believe we have adequately addressed all the excellent advices and questions raised by reviewers. Furthermore, we checked the manuscript and made sure the submitted manuscript is correct.

Please contact us if any further questions remain.

Sincerely yours,

Prof.  Chih-Cheng Chen 

Response to the comments of the reviewer:

Reviewer 2:

Comments and Suggestions for Authors

Q1: In this work, the authors investigate the development of a novel approach to detecting duplication of copyright images using a convolutional neural network architecture. While the problem is of interest, there are a few concerns regarding the soundness of the presented solution for solving the problem that must be addressed. It seems to the reviewer that the developed approach is a manipulation detector as opposed to a unauthorized copyrighted image detector. How do the authors discriminate between an authorized/copyrighted image that has one of these effects applied to it vs. a manipulation on the original image. For example, if the original copyrighted image is vertically flipped and the manipulation was to put it right side up, would this approach detect this manipulation? The authors need to better explain their methodology between what is considered an authorized image vs. an unauthorized image.

Response: Many thanks for the reviewer’s valuable comments. The definition of image copy detection is given by Wan et al.’s [8]. When image owners worry their images have been illegally manipulated and circulated over the Internet, image owners could generate various manipulation images and then feed the manipulation images and original image into the image copy detection scheme for extracting image features. Later, the image owner collects similar images from the Internet by using a search engine, i.e. Chrome, and then flitter the most suspicious images also called copy images by using image copy detection schemes. The image copy detection scheme only identifies suspicious images for the given query image. Image copy detection schemes could not determine whether the manipulation is caused by the image owner or unauthorized users. However, once the copy images have been determined, image owners shall know which ones are generated by themselves based on the source of the identified copy images. As for the rest identified copy images will be judged as illegally manipulated by unauthorized users and then image owners could claim their ownerships to those unauthorized users, and ask them to pay the penalty or take legal responsibilities. The related descriptions have been added in our revision to comply with the referee’s comment. Please refer to the lines from 56 to 64 on page 2 and the lines from 244 to 250 on page 7.

Other comments:

Q2. It would be helpful to give a direct example of the need for copyright detection technology. What is a real world problem that this solution would help solve?

Response: Many thanks for your valuable comments. A real-world application has been added in our revision to comply with the referee’s comments. Please refer to the lines from 56 to 64 on page 2 and the lines from 244 to 250 on page 7.

Q3. Watermarks are discussed as one primary way to copyright an image but wasn't considered in the permutation set by the authors.

Response: Many thanks for your detailed review. Sure, watermarking is one of the approaches to protect the copyright of images. The reason we did not test the watermarked image because if an original image has carried a watermark generated by its image owner; the corresponding copy images will also carry the hidden watermark. Therefore, we only focus on identifying the manipulation images instead of manipulated watermarked images. It is noted such an arrangement is also based on previous works [11-14]. To give readers a clear picture about our experiments, we added some sentences to explain why we do not include the watermarked image in the manipulation demonstrated in Figure 6. Please refer to the lines from 318 to 324 on page 11.   

Q4. How did the authors balance the training dataset between unauthorized image manipulations and the original images to avoid training bias?

Response: Many thanks for your detailed review. In our proposed scheme, the original image and its manipulations are not grouped as two labels, respectively. In contrast, the original image and its 43 manipulation images are formed as a group and marked as the same label. This is because our proposed image copy detection scheme aims to shrink the training time and scale of the training dataset. In other words, when the image owner wants to verify whether the similar images collected from the Internet contain her/his image. Image owners only need to generate various manipulation images and feed them into our proposed image copy detection scheme for training the features of her/his image. Then, the image copy detection scheme would identify the most suspicious image from similar images. In this revision, we have added the related statements to comply with the referee’s comment. Please refer to the lines from 244 to 250 on page 7.

  1. Doesn't seem like the authors utilized the verification dataset to avoid overfitting (e.g. through the use of 'patience' or 'early stopping' mechanisms)

Response: Many thanks for your kind reminder. In this revision, we have added the statements regarding the mechanism about avoiding overfitting in this revision to comply with the referee’s comments. It is noted that we set the default training_steps as 4000 to prevent the overfitting problem. This is because we found when the training_steps are more than 4000 it resulted in better training accuracy but lower validation accuracy. Please refer to lines 302 and 303 on page 10.

6.Very hard for the reader to see some of the manipulations shown in Figure 6.

Response: Many thanks for your detailed review. In this revision, we have enlarged the manipulations shown in Figure 6.

Round 2

Reviewer 2 Report

The authors do a good job of clarifying my original concerns with the manuscript. I think a solid grammar/English polish pass is required before acceptance but otherwise I am content with the updated version. Thanks to the authors for making these updates and providing a detailed response.

Author Response

Journal: Electronics (ISSN 2079-9292)

Manuscript ID: electronics-996564

Type: Article   Number of Pages: 16

Title: Content-based image copy detection using Inception convolutional neural network

Authors:Xiaolong Liu , Jinchao Liang , Zi-Yi Wang , Yi-Te Tsai , Chia-Chen Lin * , Chih-Cheng Chen *

Dear Editor,

Thank you very much for your letter and for the comments by the reviewers. These comments are very valuable and helpful for our paper.

We appreciate the careful, constructive, and generally favorable reviews given to our paper by the reviewers.

We believe we have adequately addressed all the excellent advices and questions raised by reviewers. Furthermore, we checked the manuscript and made sure the submitted manuscript is correct.

Please contact us if any further questions remain.

Sincerely yours,

Prof.  Chih-Cheng Chen 

  1. Why did the authors include the word 'inception' in the title? Because the inception is one of CNN structure, I think it had better exclude it in the title.

Response: Many thanks for your detailed review. In this revision, “inception” has been excluded from the title to comply with referee’s comment.

  1. In Table 5, we can see that the dataset only including 45 degree rotation cannot deal with other manipulation, such as 67, 90, 180, 270 degree rotations and crayon effect. It is not a sufficient experiment to show practicableness. It would be better to include other experiments with other manipulated images.

Response: Many thanks for your kind reminder. In this revision, we already added another experiment. Table 6 demonstrates the detection capability of our proposed scheme is increased when the training dataset includes the original images and images with 45° rotation/90° rotation. In the future, we will focus on manipulation with “Crayon Effect” to increase the detection capability of our proposed scheme.

Table 6. Comparison II of the detection results in different image processing manipulations.

Image Processing Manipulation

Proposed Scheme with data set including original images and images with 45° rotation/90° rotation

Proposed Scheme with data set including all manipulated images

Lin et al.’s Scheme

Wang et al.’s Scheme

Kim et al.’s Scheme

22.5°Rotation

Yes

Yes

Yes

No

No

45°Rotation

Yes

Yes

No(10/10)

No

No

67°Rotation

No (1/10)

Yes

Yes

No

No

90°Rotation

Yes

Yes

Yes

Yes

No

270°Rotation

Yes

Yes

Yes

Yes

No

180°Rotation

Yes

Yes

Yes

Yes

Yes

Upper Left Truncated

Yes

Yes

Yes

No

No

Upper Right Truncated

Yes

Yes

Yes

No

No

Lower Left Truncated

Yes

Yes

Yes

No

No

Lower Right Truncated

Yes

Yes

Yes

No

No

Crayon Effect

No(4/10)

Yes

No (10/10)

Yes

Yes

Mosaic

Yes

Yes

No (10/10)

Yes

Yes

Twisted

Yes

Yes

Yes

Yes

No

Central Rotation

Yes

Yes

No (10/10)

No

No

Aspect Ratio Change

Yes

Yes

Yes

No

No

Histogram Equalized

Yes

Yes

Yes

No

No

Sharpened

Yes

Yes

No (10/10)

No

No

Horizontally Shifted

Yes

Yes

Yes

No

No

Vertically Shifted

Yes

Yes

Yes

No

No

The rest 25 manipulations

Yes

Yes

Yes

Yes

Yes

Detection number

42

44

39

31

28

Detection Rate

95.45%

100%

88.6%

70.5%

63.6%

Table 6 demonstrates comparison II when images with 45° rotation/90° rotation are added to the training dataset. Here, we can see the detection performance of our proposed scheme on manipulation with different rotation angles is increased although “Crayon Effect” manipulation is still the weakness of our proposed scheme. After carefully observe the “Crayon Effect” manipulation, we found it leads the detection capability decreased when the texture of the image is more blurred.
